# Dragonflies (Odonata) in Cocoa Growing Areas in the Atlantic Forest: Taxonomic Diversity and Relationships with Environmental and Spatial Variables

**Laís R. Santos** [1,2,3] and **Marciel E. Rodrigues** [1,2,4,*]

1  Laboratory of Aquatic Organisms ("LOA"), Departament of Biological Sciences, Santa Cruz State University (UESC), Ilhéus 45662-900, Bahia, Brazil

2  Postgraduate Program in Tropical Aquatic Systems ("PPGSAT"), Department of Biological Sciences, Santa Cruz State University (UESC), Ilhéus 45662-900, Bahia, Brazil

3  Postgraduate Program in Ecology and Biodiversity Conservation ("PPGECB"), Department of Biological Sciences, Santa Cruz State University (UESC), Ilhéus 45662-900, Bahia, Brazil

4  Department of Exact and Technological Sciences, State University of Southwest Bahia (UESB), Vitória da Conquista 45083-900, Bahia, Brazil

*  Correspondence: merodrigues@uesc.br

**Abstract:** In the south of Bahia state, a large part of the native Atlantic Forest areas has been modified for the cultivation of cocoa (*Theobroma cacao*). These crops are cultivated under the shade of the canopy of native trees, a system locally known as the "cabruca" agroforestry system. This study aimed to evaluate the relationship of Odonata assemblages (adults and larvae) in cocoa farming areas and to identify the relationships of these species with local and spatial environmental variables of the monitored sites. Altogether, adult and larvae were sampled at 22 sites. Physical and physicochemical water variables were recorded for each site. A total of 1336 dragonflies were collected, of which 20 were Zygoptera species and 30 were adult Anisoptera representatives. The different life stages were related to environmental variables such as conductivity, watercourse channel width, and dissolved oxygen. The space predictors were also associated with the assemblages, mainly for adults. The present study identified that cabruca areas maintain a great diversity of dragonflies, including species that are considered to be forest specialists and more sensitive to landscape changes. The characteristics of this cropping system are considered to be favorable for the conservation of the biodiversity of the Atlantic Forest.

**Keywords:** dragonflies; land uses; Atlantic Forest; *Theobroma cacao*; cabruca

## 1. Introduction

Different land uses modify natural environments and negatively impact terrestrial and aquatic ecosystems, along with the associated biodiversity [1–3]. Among the different land uses, the conversion of forests into farmland and pastures has increased the degradation of aquatic ecosystems and water quality due to the removal of vegetation from the surrounding water bodies, silting of channels, and water pollution [3–6]. These impacts modify environments and can affect insect assemblages, thus, causing highly negative effects on aquatic ecosystems and their biodiversity [3,6,7].

Agriculture is one of the leading economic activities in Brazil. Additionally, this activity causes the highest rate of conversion of natural landscapes [3]. According to MapBiomas [8], 30.97% of Brazilian territory is used for agriculture, especially in the Pampas, Cerrado, and Atlantic Forest domains, where large areas have been converted to grow crops and graze livestock. In Atlantic Forest fragments, present in 17 Brazilian states, intensive exploitation has led to drastic reductions and modifications of native areas [3,9], and therefore, these Atlantic Forest fragments have been identified among the priority

regions for conservation. The Atlantic Forest houses vast biological diversity and is highly susceptible to human intervention [10,11].

In this regard, the southern region of the state of Bahia can be highlighted, due to the cultivation of cocoa (*Theobroma cacao* L. 1753) since the 18th century. This type of cultivation has had an important economic impact on the region. Studies in cocoa growing areas have demonstrated that these areas have been able to maintain part of the region's flora and fauna, helping to conserve biodiversity [12–14]. In the agroforestry system used to cultivate cocoa, plants grow under the shade of the native trees in the forest, locally known as cabruca or cacao-cabruca areas [13–15]. This agroforestry system is considered to be favorable for the conservation of natural resources and local biodiversity, since it maintains part of the native forest structure and, consequently, protects terrestrial ecosystems, sustains some ecological services, and maintains the species diversity of fauna and flora [12–15].

Environmental changes caused by different land uses are directly related to the structuring and maintenance of biodiversity associated with aquatic ecosystems. Since any modification to the physical conditions of surrounding water bodies alters these environments and the physicochemical parameters of the water, local assemblages are also drastically modified [16–19]. These changes in the abiotic parameters of the water, caused by the reduced riparian vegetation, increase the entry of allochthonous material and sunlight [20–22]. Therefore, it is critical to understand the relationships of environmental and spatial variables among cacao-cabruca areas and the diversity of associated species, and how this type of cultivation affects biodiversity. This understanding can help to identify and to quantify the impacts (negative and positive) of the management and use of land, thus, supporting more effective decision making to protect species and their habitats [19,21].

Among aquatic invertebrate species, dragonflies have been widely used in studies to assess the effect of changes in environmental variables on their biodiversity in areas with different land uses [20,23–26]. This is due to the fact that there are species that can, more or less, tolerate changes in the natural environment, and therefore, are able to reflect the local conditions, and therefore, have been widely used as bioindicators of water quality and the impacts caused to the surroundings of aquatic ecosystems [27–30]. Dragonflies are organisms that are extremely dependent on aquatic ecosystems, using these environments for oviposition and development of larvae, in addition to depending on the surrounding terrestrial ecosystems, since adults use these environments to feed, to defend their territories for reproduction, and to perform important physiological functions such as thermoregulation [30–32].

These ecological and behavioral characteristics of the species allow them to reflect the characteristics and integrity of the ecosystems they live in [27,30–32], which makes the different species ideal as forest and open-area specialists and habitat generalists [33]. In this regard, the relationships among the different local and spatial environmental variables and the different Odonata species in cacao-cabruca areas should be evaluated to understand the effects of this type of cultivation on Odonata biodiversity. Moreover, the group can be used as a "surrogate" for other groups of aquatic insects [34].

Accordingly, in the present study, we evaluated the diversity of Odonata in cocoa cultivation areas and the relationships with local and spatial environmental variables between adults and larvae at the sampling sites. The following two specific objectives were proposed:

(I). To assess the diversity (richness and abundance) of adult and larvae dragonflies in cocoa farming areas among sampling sites. Our prediction is that cocoa farming areas, which are considered to be agroforestry systems, can maintain a wide range of Odonata species by also maintaining groups of species considered to be forest specialists, habitat generalists, and open-area specialists [14].

(II). To assess the influence of environmental physical variables, the physicochemical variables of water, and the spatial relationships in the structuring of dragonfly assemblages (adult and larvae individuals) in cocoa farms. In the present study, the local and physical variables such as canopy cover, channel width, bank structure,

and amount of riparian vegetation sites, as well as abiotic water variables (dissolved oxygen, pH, and conductivity) and the distances between sampling sites were found to be important factors in the structuring of Odonata assemblages.

## 2. Materials and Methods

### 2.1. Study Area

The southern coastal area of the state of Bahia is inserted in the central corridor of the Atlantic Forest domain, between the south of Bahia and the north of Espírito Santo, which comprises coastal tablelands or tableland forests in a subunit of dense ombrophilous forest [35]. This region has high species diversity and some degree of endemism [35]. Agriculture is the main economic activity in the region [36] including cocoa cultivation, which is one of the five most relevant permanent crops. According to Köppen-Geiger, the climate in the region falls in the tropical rainforest Af classification (tropical super humid), with rainfall evenly distributed throughout the year.

This study was conducted in six municipalities, namely Una, Buerarema, São José da Vitória, Ilhéus, Uruçuca, and Itacaré, which make up the southern region of the state of Bahia. The adult specimens were collected in 22 streams, 16 of which were also used to collect the larval specimens (Figure 1) (Appendix A, Table A1). The samples were collected in first- to third-order streams from September to November 2019 and in July and August 2020. The selected properties for sampling belonged to organic cocoa producers of the Cooperativa Cabruca (an agricultural cooperative). In these areas, the cocoa plants are grown under the shade of native trees and in consortium with others crops, such as banana, açai berry, cupuaçu, vanilla, and palm oil. The properties are considered to be small with an organic cultivation system and without the use of pesticides. The channels of the lotic environments sampled were mostly located in the middle of the cocoa growing areas.

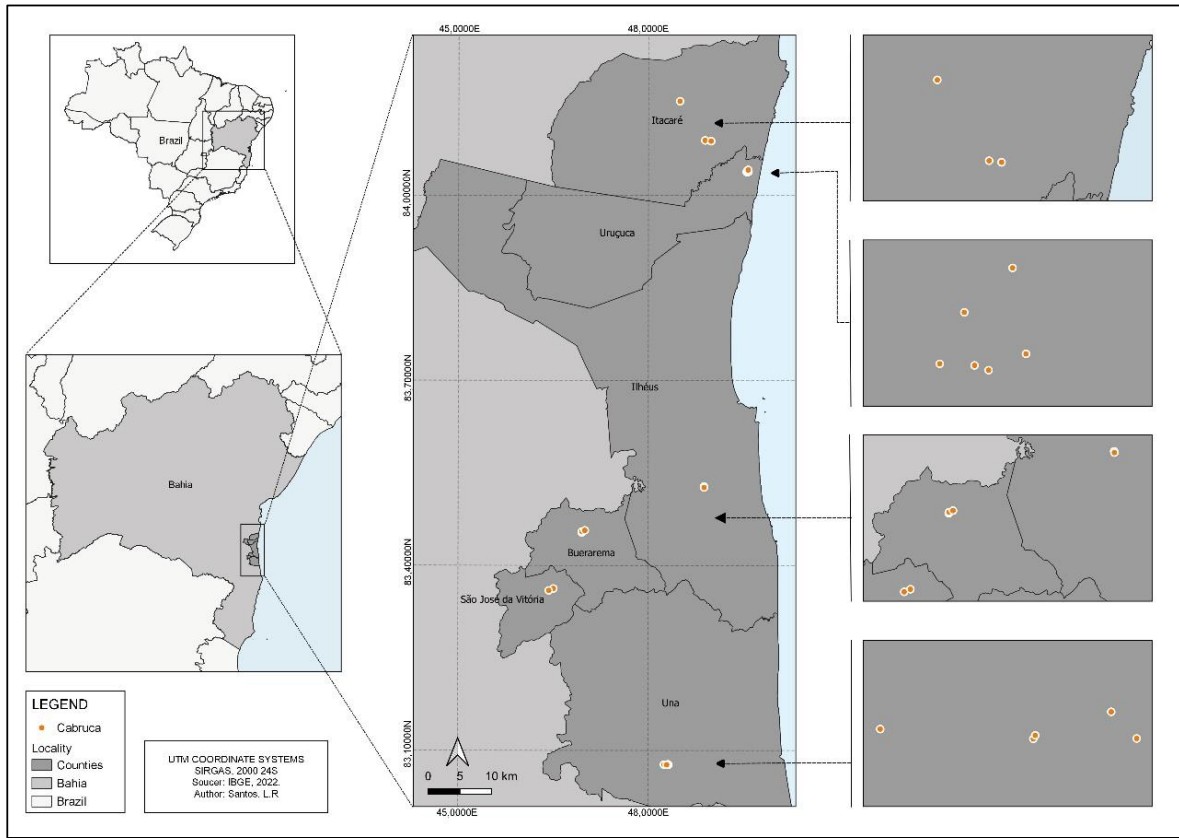

**Figure 1.** Map with sites sampled in the six municipalities in the southern region of Bahia in cabruca areas. The red circular dots are the sites sampled.

*2.2. Sample Collection*

The adult specimens were collected with an entomological net, in segments of 100 m on both banks of the streams using an entomological net and a sampling effort of 01:30 h for each sampled stream between 9:00 am and 4:00 pm. Two sampling campaigns were carried out at each stream to increase the representativeness of the assemblages at each sampling site.

The larvae were collected in areas of rapids and backwater in different habitats using a D-frame net and sieve, in all compartments of the environment. A 30 m segment on the shore was selected along each stream and 20 stretches of 1 m were sampled in proportion to the number of registered habitats in each stream (sand, leaves, particulate organic matter, gravel, consolidated bedrock), according to the method of Barbour [37]. Active searches were performed using the sieve, for 15 min, in the different habitats within the 100-m stretch in areas that had not been sampled with the D-frame net.

Subsequently, the specimens were sent to the Laboratory of Aquatic Organisms ("LOA") of the Santa Cruz State University (UESC) for curation and identification based on available keys [23,38–42]. The collected material is currently deposited in the Aquatic Insect Collection of UESC.

*2.3. Environmental and Spatial Variables*

2.3.1. Physical Variables of the Channels and Surroundings and Physicochemical Variables of the Water

The environmental variables were measured simultaneously with sample collection at each sampling point, where the physical structure of the habitats was evaluated using the habitat integrity index (HII), as proposed by Nessimian [43]. The HII is based on 12 habitat characteristics that evaluate the stream structure in relation to the characteristics of the forest areas, the land use pattern, retention mechanisms, aquatic vegetation, substrate, and debris. Finally, the obtained values of each metric were transformed using the method of Oliveira-Junior and Juen [27]. The sum of the 12 qualitative habitat characteristics varies from 0.1 to 1, where the lowest values are characteristic of locations with lower environmental integrity and values close to 1 refer to more intact or preserved locations. The HII has been widely used in ecological studies to evaluate the integrity of aquatic ecosystems and it is considered to be a good tool in studies that evaluate these changes in aquatic ecosystems [17].

In order to evaluate the amount of canopy coverage, three photographic images were obtained in the center of the stream every 10 m within the 30 m segment. In each of these segments, a photograph of the canopy was taken with a mobile phone camera positioned 30 cm over the water level, for a total of three images for every sampled stream. The images were uploaded for treatment with the Image J software to calculate the light and dark areas of each image. The software produces a black and white binary image of the photographs that differentiates the regions with light input from the canopy cover. The generated image was used to calculate the white area, which corresponds to the area of light input. After performing this process on the three images, the average light input between the images was calculated for each stream.

To obtain local riparian vegetation integrity data on both banks of the streams, three quadrants of 100 m$^2$ (10 m × 10 m) were selected along the 30 m stretch on each bank. Of these six quadrants, three quadrants were randomly selected to record the total number of native trees with more than 15 cm of circumference at breast height (CBH), followed by a calculation of the diameter at breast height (DBH). The number of cocoa individuals within each of the quadrants was also counted. In the field, we observed that this information could provide valuable insight into the relationships of Odonata assemblages at the sampling points.

The amount of forest cover was calculated from the information on the coordinates taken from each of the points sampled with the aid of GPS (Etrex 10). This information was loaded into the Qgis desktop software 3.16.9 Geographic Information System.

The images used to extract forest cover data were obtained from the MapBiomas land cover and use database, with a spatial resolution of 30 m (30 × 30). After the delimitation of the cultivation areas, buffers with a radius of 1000 m were created around the sampled points and then, the amount of coverage within each buffer was calculated (Appendix A, Table A1).

The width, depth, and current speed of the stream were measured along the 30 m stretch, totaling five measurements per point. A multiparameter probe (YSI Professional) was used to collect the physicochemical parameters of the water (temperature, pH, conductivity, dissolved oxygen (OD), and salinity).

All the values of the collected variables used in the analyses represent the means between the two sampling campaigns and the number of replicas at each point (Appendix A, Table A2). These variables were used as environmental predictors given their critical importance in the structuring of aquatic insect assemblages [20,26,34,44].

### 2.3.2. Spatial Variables

Spatial data were initially collected by calculating spatial filters from the geographical coordinates of each sampled stream. The coordinates were captured using a Garmin eTrex GPS. These coordinates were used to construct a Euclidean distance matrix using the "vegdist" function of the "vegan" package [45]. The spatial filters were calculated by analyzing the principal coordinates of neighbor matrices (PCNM), which, in turn, were also calculated using the "vegan" package with the "pcnm" function. This method can determine whether spatial predictors are structuring the distribution of assemblages [19]. The significant axes were selected using the "adespatial" package [45] in the R software. The PCNM vectors were submitted to the forward selection model [46]. The selected vectors were subsequently used in the redundancy analysis (RDA). All the methods mentioned above were used to test the influence of spatial variables on adult and larvae specimens.

### 2.3.3. Data Analysis

The data matrix containing the abundance of adult and larvae specimens was transformed, separately, into a relative abundance matrix using Hellinger's method to reduce the effect of large abundances [46]. Each stream was considered to be a sample unit, totaling 22 for adults and 16 for larvae. The relationship of group diversity with the with amount of forest cover was assessed by sorting the relative abundance of the adult species and the genera of the larvae with the amount of forest cover of the sampled areas using the "generic" function of R.

The local physical and physicochemical variables of the water were subjected to Spearman correlation analysis to exclude correlated variables (>70%). None of these variables presented this level of correlation, so none were excluded. Subsequently, these variables were used to perform a principal component analysis (PCA) using Euclidean distance to evaluate their relationship with the sampling sites. The axes of the PCA were tested with the broken stick method [47,48] to determine which axes would be used as environmental variables, and later in the analysis of the adult and larvae of the species collected.

The combinations between spatial and environmental predictors and the composition of adult and larvae assemblages were analyzed separately. The selected spatial and environmental vectors were used in the redundancy analysis (RDA) for each of the different life stages (adult and larvae). The RDA significance was tested by analysis of variance (ANOVA). All analyses were performed using the R software, with the "vegan" package [49] and "adespatial" package [45].

### 3. Results

A total of 689 adult Odonata individuals were captured, representing seven families and 50 species, 520 individuals distributed in 20 species of Zygoptera and 169 individuals and 30 species of Anisoptera (Appendix A, Table A2. Among the Zygoptera, the most

representative species were *Argia chapadae* Calvert, 1909; *Hetaerina rosea* Selys, 1853; and *Heteragrion lencionii* Vilela, Farias & Santos, 2021, with, respectively, 154, 87, and 87 individuals. Among the Anisoptera, *Erythrodiplax fusca* Rambur, 1842, and *Perithemis thais* Kirby, 1889 totaled 62 and 18 individuals, respectively.

In relation to larvae, 647 specimens were collected, comprising five families and 11 genera of Zygoptera and three families and 23 genera of Anisoptera. The most representative genera were *Homeoura* and *Heteragrion* with 43 and 76 individuals, respectively, (Zygoptera), along with *Epigomphus* with 72 individuals (Anisoptera). Regarding the life stages (adult and larvae), 14 genera were collected, four genera of the suborder Zygoptera and 10 genera of the suborder Anisoptera. Sixteen genera were only found as adults and 20 genera were recorded only for the larvae (Appendix A, Tables A3 and A4).

The relationship between the relative abundance with the amount of forest cover in the cocoa areas indicated alteration of some species as a function of the amount of forest cover for adult and larvae in the evaluated areas (Figures 2 and 3). Some species and genera were related to areas with low amounts of forest cover in the sampled areas. The most open-area specialist genera and species targeted sites with a lower amount of forest cover, as in the case of the species *Telebasis willinki* Fraser, 1948; *Acanthagrion aepiolum* Tennessen, 2004; *Erythrodiplax famula* Erichson in Schomburgk, 1848; *Elasmothemis alcebiadesi* Santos, 1945; and *Diastatops nigra* Montgomery, 1940; and the genera *Castoraeschna*, *Roppaneura*, *Erythrodiplax*, *Enallagma*, *Lestes*, and *Tramea*, located on the left side of both graphs (Figures 2 and 3).

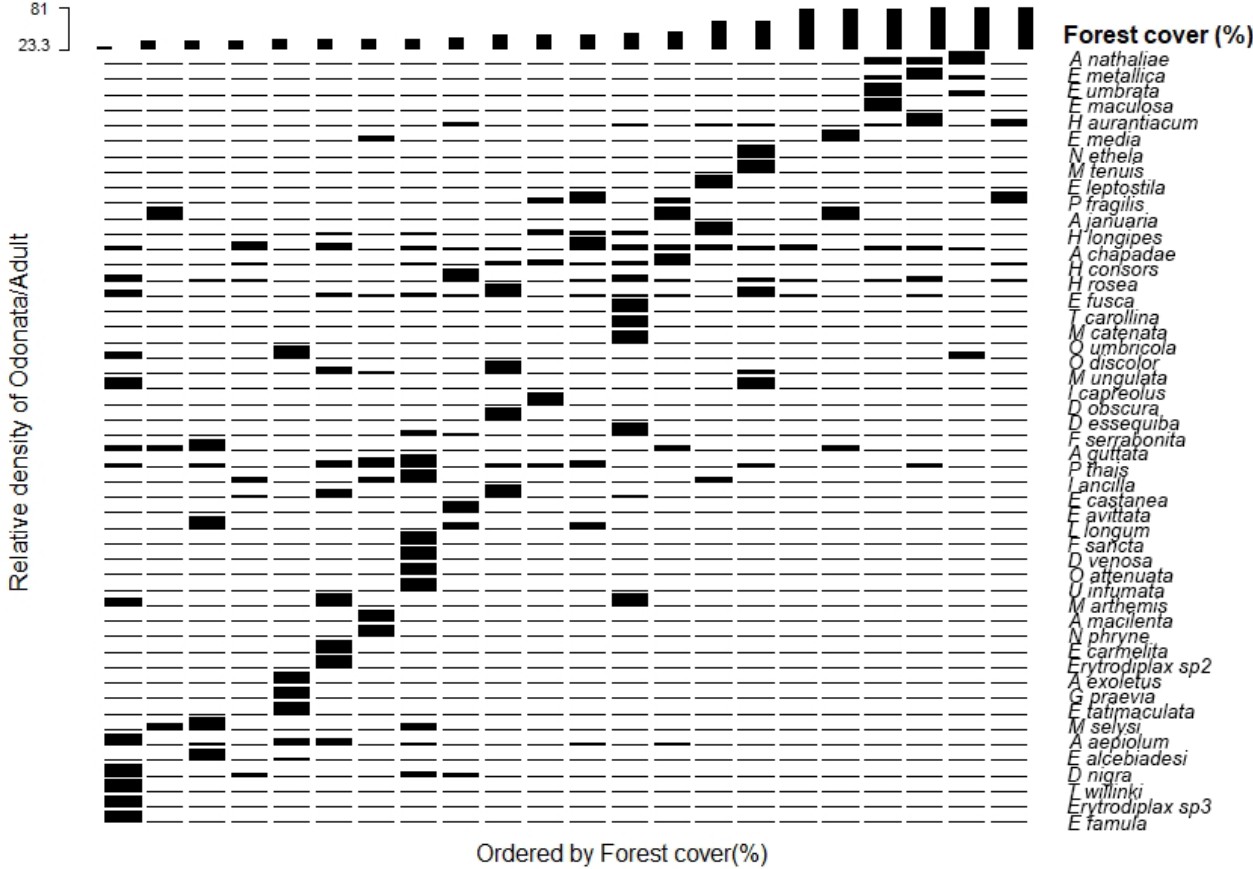

**Figure 2.** Relative abundance of adult individuals of Odonata with the amount of forest cover (%) for the streams sampled in cabruca areas.

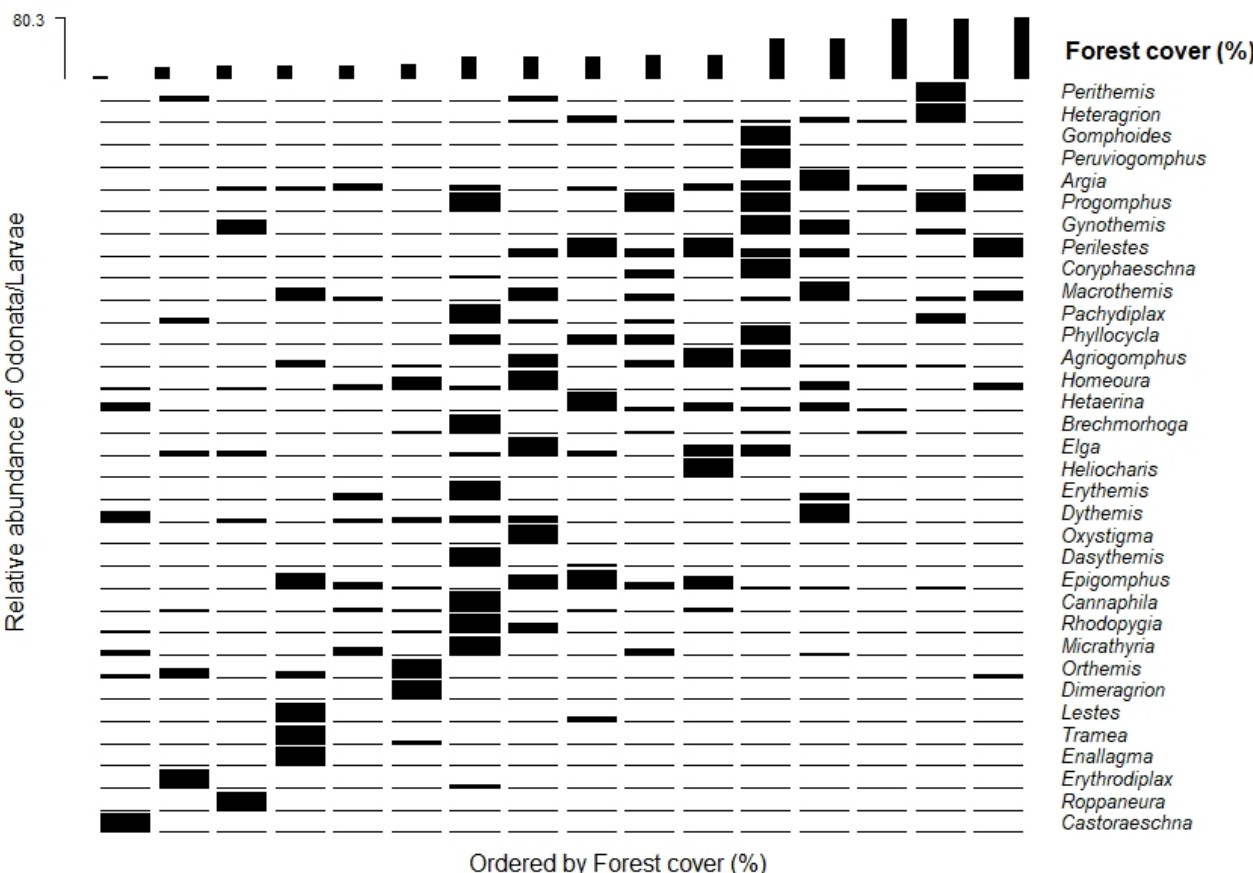

**Figure 3.** Relative abundance of larvae of Odonata with the amount of forest cover (%) for the streams sampled in cabruca areas.

The species of adults and genera of larvae considered to be habitat generalists are represented in the central strips of the charts, which show that these groups may have greater tolerance for the loss of forest cover surrounding streams in the cocoa growing areas (Figures 3 and 4). Finally, species and genera that were associated with cocoa growing areas with greater amounts of forest cover were *Aceratobasis nathaliae* Lencioni, 2004; *Epipleoneura metallica* Rácenis, 1955; *Erythrodiplax maculosa* Hagen, 1861; and *Heteragrion aurantiacum* Selys, 1862. The genera *Peruviogomphus*, *Gomphoide*, and *Heteragrion* were more abundant in the cacao-cabruca areas with a higher level of amount forest cover (Figure 4).

The environmental variables in the PCA analysis for adults presented the first four axes and the observed values were greater than those estimated by the broken stick method. The four axes of the PCA accounted for 58.48% of the cumulative proportion. The PC1 axis explained 17.47%, PC2 15.11%, PC3 14.53%, and PC4 11.37% (the eigenvalues for the axes were PC1 = 4.541, PC2 = 3.927, PC3 = 3.777, and PC4 = 2.957). The variables that contributed the most on each axis were greater than 0.6. For the PC1 axis, the variables were conductivity (−0.684), salinity (−0.758), and margin structure (0.630). For the PC2 axis, it was stream width (0.665). In the formation of the PC3 axis, the variables were depth (−0.679), undercut margin (0.630), aquatic vegetation (0.614), and HII (0.727). For the PC4 axis, the variables were dissolved oxygen (0.785) and luminosity (0.686).

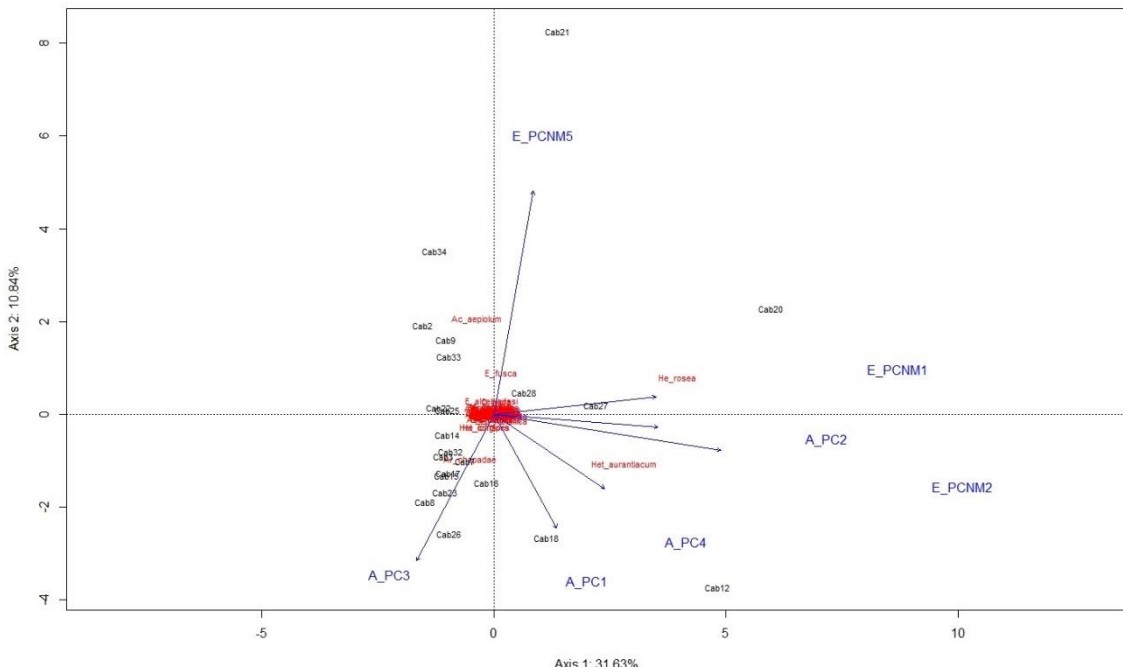

**Figure 4.** Result of the redundancy analysis showing the relationships among spatial variables and environmental variables with the sample sites and adult individuals of Odonata. A _= Environmental variable axes and E_ = Spatial variable axes.

For the larvae, the first three axes showed the best response in the PCA analysis according to the broken stick method. These accounted for 50.64% of the cumulative PCA proportion. The PCA1 axis explained 20.32%, PCA2 15.67%, and PC3 14.65% (respectively, the eigenvalues for the axes were 5.283; 4.073, and 3.807). The environmental variables that most influenced the formation of axes with values greater than 0.6 included conductivity (−0.639), dissolved oxygen (−0.747), salinity (−0.678), and margin structure (0.669) for PCA1, width (−0.787), channel/sediment (0.603), and current and backwater, or meanders (−0.646) for PCA2, and riparian forest width (0.621) and habitat integrity index (0.608) for PCA3.

For the spatial variables, the PCNM axes 1, 2, and 5 were selected by the forward selection method, corresponding to the axes with better correlation with the data when related to adult individuals. The obtained results were PCNM2 with correlation factor $R^2 = 0.199$ (F = 4.97, $p = 0.006$); PCNM1 with $R^2 = 0.116$ (F = 3.24, $p = 0.014$), and PCNM5 with $R^2 = 0.084$ (F= 0.08, $p = 0.018$). For the case of larvae, the selected axes were PCNM2 and -1. The obtained results were PCNM2 with R = 0.200 (F = 3.493, $p = 0.006$) and PCNM1 with $R^2 = 0.116$ (F= 2.205, $p= 0.030$) (Table 1).

**Table 1.** PCNM axes of spatial predictors selected by the forward selection model for adult and larvae among the streams in cocoa areas in the southern region of the state of Bahia.

|  |  | Axes | Order | $R^2$ | $R^2$Cum | AdjR$^2$Cum | F | $p$ Value |
|---|---|---|---|---|---|---|---|---|
|  | 1 | PCNM2 | 2 | 0.199 | 0.199 | 0.159 | 4.979 | 0.006 |
| Adult | 2 | PCNM1 | 1 | 0.116 | 0.316 | 0.244 | 3.245 | 0.014 |
|  | 3 | PCNM5 | 5 | 0.084 | 0.400 | 0.300 | 2.536 | 0.018 |
| Larvae | 1 | PCNM4 | 4 | 0.235 | 0.235 | 0.195 | 5.867 | 0.038 |
|  | 2 | PCNM1 | 1 | 0.114 | 0.360 | 0.278 | 3.177 | 0.026 |

The result of the redundancy analysis (RDA) indicated that the combination of environmental and spatial variables accounted for 42.74% of the assemblage composition of adult individuals of Odonata in the cabruca areas. Axis 1 explained 31.63% and Axis 2

explained 10.84% of the data variation (Figure 4). The ANOVA showed that the ordering analysis generated by the RDA was statistically significant (F = 2.67, *p* = 0.001). The points most influenced by the evaluated axes were Sites 18 by Axis 1 of the PCA, Sites 8, 14, 15, 16, 17, 23, 26, and 32 by Axis 3 of the PCA, Sites 20, 27, and 28 by Axis 1 of the spatial analysis, and Sites 2, 9, 21, 33, and 34 by Axis 5 of the spatial analysis. The species associated with environmental and spatial variables were *Acanthagrion aepiolum* and *Erythrodiplax fusca* for the spatial axis (PCNM5) and *Hetearina rosea* for the spatial axis (PCNM1). The species *Heteragrion aurantiacum* was associated with the PC4 axis of the environmental variables and the species *Argia chapadae* with the PC3 axis. The partition analysis of the environmental and spatial variables indicated that the environmental variables accounted for 32.75%, the spatial variable for 40.06%, and the two together in the model explained 57.25% of the data variation (Table 2).

**Table 2.** Partition of the RDA for environmental and spatial variables related to adult individuals.

| Variables | Df | $R^2$ | Adj.R.Squared |
|---|---|---|---|
| Environment | 4 | 0.32753 | 0.16930 |
| Spatial | 3 | 0.40061 | 0.30071 |
| Environment + spatial | 7 | 0.57255 | 0.35883 |
| Residue | | | 0.64117 |

The RDA performed with the assemblages of larvae of Odonata and the environmental and spatial variables resulted in 31.58% of the explanation obtained in the first four axes that contributed the most (Axis 1, 12.40% and Axis 2, 7.83%). The ranking generated by ANOVA did not show significant values (F = 1.269, *p* = 0.059) (Figure 5). However, from the graph, it was observed that the sites associated with environmental and spatial variables were Sites 7, 17 and 28 for the PC1 axis, Site 2 for the PC2 axis, Sites 27 and 34 for the axis PC3, and Site 16 for the PCNM2 axis. The partition analysis indicated that the environmental variable was responsible for 27%, the spatial variable for 19%, and both explained 36%; the variables are presented in Table 3.

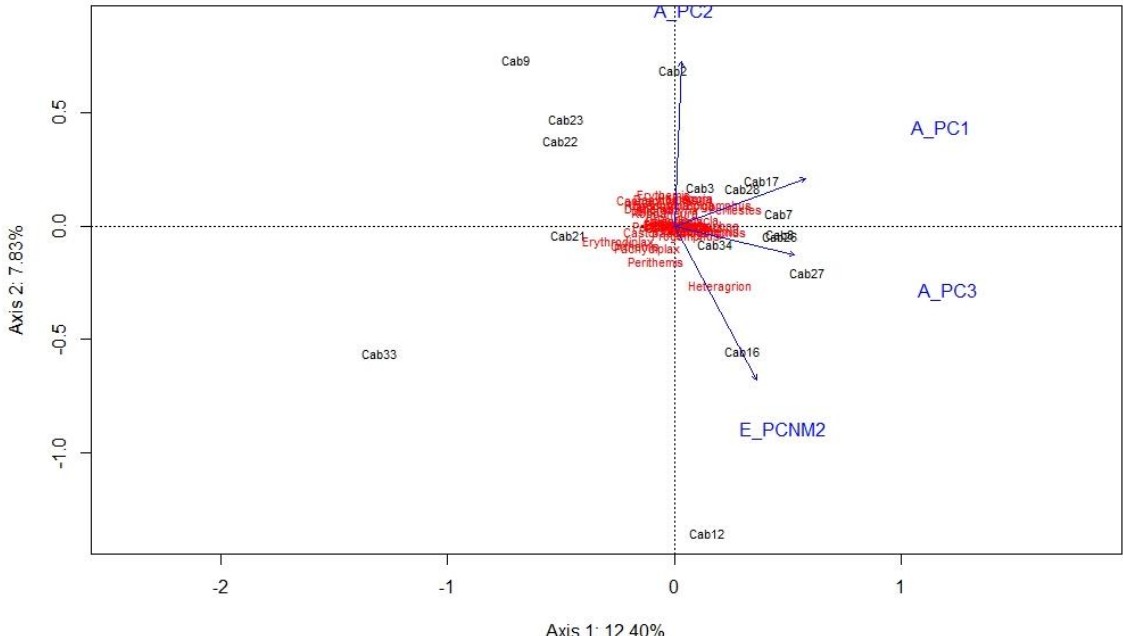

**Figure 5.** Results of the redundancy analysis showing the relationships among spatial variables, environmental variables, and larvae of Odonata. A_ = Environmental variable axes and E_ = Spatial variable axes.

**Table 3.** Partition of the RDA for environmental and spatial variables related to larvae.

| Variables | Df | $R^2$ | Adj.R.Squared |
|---|---|---|---|
| Environmental | 3 | 0.27670 | 0.09588 |
| Spatial | 1 | 0.19581 | 0.13836 |
| Environmental + Spatial | 4 | 0.36794 | 0.13810 |
| Residue | | | 0.86190 |

## 4. Discussion

The results indicated that the cabruca areas maintain a vast diversity and richness of Odonata species (adults and larvae). Studies with the group in the Atlantic Forest that have compared different land uses have highlighted the importance of cocoa cultivation areas for the diversity of species in the region [14]. Notably, these areas contain species that are considered to be specialists of forests areas, such as *Epipleoneura metallica* Rácenis, 1955; *Aceratobasis nathaliae* Lencioni, 2004; *Heteragrion aurantiacum* Selys, 1862; *Forcepsioneura serrabonita* Pinto & Kompier, 2018; and *Perilestes fragilis* Hagen in Selys, 1862. According to the data, the distribution of these species in cocoa farms was related to greater environmental integrity of the sampled areas, which directly reflected the local environmental variables. In other words, the environmental and spatial variables in the cabruca areas played important roles in structuring these Odonata assemblages. Most of the environmental variables were common to the two evaluated life stages.

Regarding the relative abundance of species with the amount of forest cover in cocoa areas, a similar pattern of occurrence was observed between adult species and larval genera. The cabruca areas with the lowest amount of forest cover values showed greater relative abundance of adult species and larval genera, formed by species considered to be specialists of open areas, associated with areas with greater canopy opening, greater solar incidence, more lentic environments, and the presence of macrophytes [50,51]. Some examples are *Erythrodiplax latimaculata* Ris, 1911; *Telebasis willinki* Fraser, 1948; *Diastatops nigra* Montgomery, 1940; and *Elasmothemis alcebiadesi* Santos, 1945 for the adults, and the genera *Erythrodiplax, Enallagma,* and *Castoraeschna* for the larvae.

A broad range of species had its distributions along the entire gradient of the forest cover, and therefore, can be considered to be habitat generalist species [50,52], as in the cases of *Hetaerina rosea* Selys, 1853; *Argia chapadae* Calvert, 1909; and *Micrathyria ungulata* Förster, 1907; and the genera *Progomphus*; *Argia*, 1842; and *Hetaerina.* The relative abundance of some adult species and genera of larvae was associated with areas with a higher amount of forest cover, as in the cases of the species *Aceratobasis nathaliae* Lencioni, 2004; *Epipleoneura metallica* Rácenis, 1955; and *Heteragrion aurantiacum* Selys, 1862, usually considered to be forest specialist species, and the genera *Heteragrion, Gomphoide*; and *Peruviogomphus*, composed of more sensitive individuals that need greater environmental integrity and conditions more comparable to forest areas [53,54]. However, it is worth mentioning that some species considered to be specialists in forests that have already been recorded in the region in areas of native forest in the Atlantic Forest [14] were not recorded in the cocoa cultivation areas sampled. For example, *Aceratobasis cornicauda* Calvert, 1909; *Heliocharis amazona* Selys, 1853; *Kiautagrion acutum* Santos, 1961; and *Leptagrion macrurum* Burmeister, 1839, the last two species being phytotelmata [14]. Emphasizing that even crops that maintain a part of the native vegetation such as cabruca areas may not maintain some groups of species considered to be more sensitive or with more specialized habitats.

The environmental variables that most contributed to the formation of the PCA axes, for adults, included conductivity, salinity, dissolved oxygen, margin structure, stream width, channel undercut margin, depth, aquatic vegetation, and luminosity. For the larvae, the variables that most contributed to the formation of the PCA axes were almost the same as for the adults, except aquatic vegetation and undercut margin. These results emphasize the importance of the same set of environmental variables for dragonfly species (adults and larvae) in cocoa growing areas, and that the changes in these variables can modify the structuring of local assemblages in cabruca areas. These environmental variables have

already been highlighted in other studies that have evaluated the structuring of Odonata assemblages in other types of land use or with different environmental modifications [20,24,26,29].

In cabruca areas, changes in the amount forest cover, riparian vegetation, canopy openings, erosion of stream banks, increase the number of vascular plants in the channels, and sediments in the stream, as well as the air and water temperature. These modifications may favor the abundance and permanence of species considered to be open-area specialists and habitat generalists [33], as in the cases of *Perithemis thais* Kirby, 1889; *Argia chapadae* Calvert, 1909; *Telebasis willinki* Fraser, 1948; and *Erythrodiplax latimaculata* Ris, 1911, which have also been recorded in other studies of altered areas [25,26,32,55,56].

Physicochemical variables of water also play important roles in structuring the assemblages of aquatic insects. Changes in water parameters and habitat affect most aquatic organisms, including Odonata [26]. The conductivity, salinity, and dissolved oxygen variables stand out in the structuring of Odonata assemblages (adults and larvae) and can serve as parameters to define the specific oviposition sites selected by adults and are closely associated with development from larvae stages of many Odonata species [42,57]. This is especially true for more sensitive species that depend on more pristine environments and more particular habitat conditions for their development. The water in more preserved environments has higher dissolved oxygen concentrations [58], which can help ensure the permanence of more sensitive species such as forest specialists. These forest specialists include *Heteragrion aurantiacum* Selys, 1862; *Forcepsioneura serrabonita* Pinto & Kompier, 2018; and *Perilestes fragilis* Hagen in Selys, 1862; as well as species of the genera *Heteragrion*, *Peruviogomphus*, and *Agriogomphus* [26,56,59–62].

Assemblage structuring between the sampling sites was also associated with the axes of spatial analysis. These results suggest that the spatial distance between the sampling points plays a critical role in the structuring of the assemblages in the cocoa cultivation sites. Areas of the same region have more homogeneous assemblages owing to the dispersal capacity of most of the sampled Odonata adults. The species *Acanthagrion aepiolum* Tennessen, 2004; *Erythrodiplax fusca* Rambur, 1842; and *Hetaerina rosea* Selys, 1853, which are open-area specialists (the first two) and habitat generalists (the third species), have also been associated with spatial filters. More generalist or open-area specialist species or species with greater dispersal capabilities are usually associated with spatial filters [26,33,51,63–65].

In general, cabruca areas are favorable environments for the conservation of habitat structure and for the colonization of more sensitive species. Moreover, such areas are considered to have sustainable agricultural cultivation, as revealed in several studies with other groups such as mammals, birds, and insects (dragonflies) [12,14,66]. However, few studies have related these areas to invertebrates in general and especially to aquatic invertebrates. In short, the results of this study emphasize that: (1) The cabruca areas manage to maintain a high diversity of Odonata and several species generally associated with more pristine areas. (2) The environmental and spatial variables are determinants in the structuring of Odonata assemblages in cabruca areas. More preserved areas or areas with higher amounts of forest cover maintain assemblages of species that are more sensitive to human impacts, whereas areas with lower amounts of forest cover have Odonata assemblages with more habitat generalist or open-area specialist species. (3) The relationship of the species with the amount of forest cover in the cabruca areas is reflected in the relative abundance of the species collected in the different sampled sites. From this perspective, the present study highlights that this cultivation system may be supporting the conservation of Odonata biodiversity in Atlantic Forest areas. Therefore, it is critical to ensure the integrity of these areas to maintain groups of more sensitive species. Notably, in Brazil, the recently approved Law 14,119 of 13 January 2021 establishes the National Policy on Payment for Environmental Services, which encourages recovery and recomposition through the planting of native species or by agroforestry systems. This law can further strengthen the relevance of cabruca areas as sites able to maintain large species diversity and as areas of economic, environmental, and social importance.

## 5. Conclusions

The present study identified that cabruca areas maintain a great diversity of dragonflies, including species that are considered forest specialists and more sensitive to landscape changes. Moreover, the local and spatial environmental characteristics proved to be important factors in the structuring of these assemblages in the cabruca areas. The characteristics of this cropping system are considered to be favorable for the conservation of the biodiversity of the Atlantic Forest. In this regard, conserving part or some of the characteristics of native habitats has contributed to maintain local species that are more sensitive to changes in natural landscapes, which confirms the importance of this agroforestry system for the conservation of dragonfly species in the Atlantic Forest areas.

**Author Contributions:** Conceptualization, M.E.R. and L.R.S.; methodology, M.E.R. and L.R.S.; formal analysis, M.E.R. and L.R.S.; investigation, M.E.R. and L.R.S.; resources, M.E.R.; data curation, M.E.R. and L.R.S.; writing—original draft preparation, M.E.R. and L.R.S.; writing—review and editing, M.E.R. and L.R.S.; supervision, M.E.R.; project administration, M.E.R.; funding acquisition, M.E.R. All authors have read and agreed to the published version of the manuscript.

**Funding:** This research was funded by Fundação de Amparo à Pesquisa do Estado da Bahia (FAPESB) for fellowships for the first author L.R.S. (process number 073.6787.2020.0007415-43). The Santa Cruz State University (UESC) and to the National Council for Scientific and Technological Development (CNPQ), for funding the research project (registered number UESC/PROPP 0220.1100.1693 and registered number CNPQ 423737/2018-0).

**Institutional Review Board Statement:** Not applicable.

**Data Availability Statement:** Not applicable.

**Acknowledgments:** The authors thank Francisco Valente Neto and Fernando Carvalho for their suggestions and criticisms of this manuscript, and the Cooperativa Cabruca for allowing research and support in field activities. Thanks are also due to the friends of the Odonatólogos da Bahia group for their support during collections, sorting, material identification, and friendship and to the post-graduate program in Tropical Aquatic Systems–PPGSAT/UESC. Thanks also to the Dean of Research and Graduate Studies (PROPP/UESC) for payment of the translation of the manuscript, and to Wagner Magalhães for reading, revising, and correcting the English in the text. To Saulo Araujo for helping with the figures.

**Conflicts of Interest:** The authors declare no conflict of interest. They also state that the funders had no role in the study design; in the collection, analysis or interpretation of data; in writing the manuscript; or in the decision to publish the results.

## Appendix A

**Table A1.** Geographical coordinates of sites sampled in cocoa-growing areas in Bahia. Total value of the habitat integrity index (HII). Information on the sampling of adults and/or larvae at each of the sites sampled.

| Points | Lat (utm) | Long (utm) | Total Value HII | Amount Forest Cover (%) | Adult Sampling | Larvae Sampling |
|--------|-----------|------------|-----------------|-------------------------|----------------|-----------------|
| Cab2 | −14.43833 | −39.04059 | 0.69 | 33.2 | X | X |
| Cab3 | −14.43848 | −39.04015 | 0.67 | 41.9 | X | X |
| Cab7 | −14.43828 | −39.04168 | 0.69 | 44.1 | X | X |
| Cab8 | −14.43667 | −39.04091 | 0.7 | 41.8 | X | X |
| Cab9 | −14.43528 | −39.0394 | 0.6 | 41.6 | X | X |
| Cab12 | −15.04876 | −39.3252 | 0.73 | 79.4 | X | X |

**Table A1.** *Cont.*

| Points | Lat (utm) | Long (utm) | Total Value HII | Amount Forest Cover (%) | Adult Sampling | Larvae Sampling |
|--------|-----------|------------|-----------------|------------------------|----------------|-----------------|
| Cab14 | −15.05157 | −39.33198 | 0.6 | 78.6 | X | - |
| Cab15 | −14.89933 | −39.10437 | 0.64 | 78.0 | X | |
| Cab16 | −14.90076 | −39.10413 | 0.72 | 78.6 | X | X |
| Cab17 | −14.33459 | −39.13932 | 0.75 | 80.3 | X | X |
| Cab18 | −14.39319 | −39.09369 | 0.76 | 81.0 | X | - |
| Cab20 | −14.39199 | −39.1025 | 0.66 | 37.0 | X | - |
| Cab21 | −14.39305 | −39.09368 | 0.67 | 23.3 | X | X |
| Cab22 | −14.96656 | −39.28364 | 0.68 | 35.1 | X | X |
| Cab23 | −14.96529 | −39.28327 | 0.6 | 32.5 | X | X |
| Cab25 | −14.96356 | −39.27904 | 0.81 | 33.6 | X | - |
| Cab26 | −14.43833 | −39.04059 | 0.8 | 60.4 | X | X |
| Cab27 | −14.43848 | −39.04015 | 0.8 | 43.7 | X | X |
| Cab28 | −14.43828 | −39.04168 | 0.7 | 60.6 | X | X |
| Cab32 | −14.43667 | −39.04091 | 0.78 | 31.0 | X | - |
| Cab33 | −14.43528 | −39.0394 | 0.64 | 31.0 | X | X |
| Cab34 | −15.04876 | −39.3252 | 0.7 | 33.0 | X | X |

**Table A2.** Measurements of local environmental variables (minimum and maximum values, average, and standard deviation) in the cacao-cabruca areas, in the streams sampled in the southern region of Bahia.

| Variables | Number of Measurements per Sites | Minimum | Maximum | Average | Standard Deviation (SD) |
|-----------|----------------------------------|---------|---------|---------|-------------------------|
| Width (cm) | 5 | 33.00 | 292.00 | 124.70 | 70.47 |
| Depth (cm) | 5 | 4.40 | 96.00 | 24.10 | 24.13 |
| Velocity (m/s) | 3 | 2.89 | 64.45 | 21.71 | 19.82 |
| Temperature (°C) | 2 | 20.00 | 24.70 | 22.22 | 1.01 |
| Conductivity | 2 | 29.10 | 90.80 | 60.45 | 21.04 |
| Ph | 2 | 5.31 | 8.91 | 6.64 | 0.94 |
| Dissolved oxygen (mg/L) | 2 | 0.53 | 65.44 | 11.24 | 15.30 |
| Salinity | 2 | 0.01 | 0.04 | 0.02 | 0.01 |
| Total native trees | 3 | 0 | 14.00 | 3.45 | 12.67 |
| Native tress CAP * (cm) | 3 | 0 | 291.00 | 116.34 | 73.44 |
| Brightness | 3 | 62,970 | 61,450,333 | 2,878,130 | 13,082,298 |

* CAP Circumference above the chest.

**Table A3.** Species recorded in the cacao areas, in the streams sampled in the southern region of Bahia.

| SUBORDER | Family/Species | Abundance |
|----------|----------------|-----------|
| ZYGOPTERA | | |
| | CALOPTERYGIDAE | |
| | *Hetaerina longipes* Hagen in Selys, 1853 | 22 |
| | *Hetaerina rosea* Selys, 1853 | 107 |
| | COENAGRIONIDAE | |
| | *Acanthagrion aepiolum* Tennessen, 2004 | 86 |

**Table A3.** *Cont.*

| SUBORDER | Family/Species | Abundance |
|---|---|---|
| ZYGOPTERA | | |
| | *Aceratobasis macilenta* Rambur, 1842 | 1 |
| | *Aceratobasis nathaliae* Lencioni, 2004 | 4 |
| | *Argia chapadae* Calvert, 1909 | 144 |
| | *Epipleoneura metallica* Rácenis, 1955 | 6 |
| | *Forcepsioneura sancta* Hagen in Selys, 1860 | 1 |
| | *Forcepsioneura serrabonita* Pinto & Kompier, 2018 | 8 |
| | *Idioneura ancilla* Selys, 1860 | 6 |
| | *Ischnura capreolus* Hagen, 1861 | 4 |
| | *Metaleptobasis selysi* Santos, 1956 | 4 |
| | *Neoneura ethela* Williamson, 1917 | 1 |
| | *Telagrion longum* Selys, 1876 | 4 |
| | *Telebasis corallina* Selys, 1876 | 2 |
| | *Telebasis willinki* Fraser, 1948 | 1 |
| | LESTIDAE | |
| | *Archilestes exoletus* Hagen in Selys, 1862 | 4 |
| | HETERAGRIONIDAE | |
| | *Heteragrion aurantiacum* Selys, 1862 | 75 |
| | *Heteragrion consors* Hagen in Selys, 1862 | 34 |
| | PERILESTIDAE | |
| | *Perilestes fragilis* Hagen in Selys, 1862 | 6 |
| ANISOPTERA | GOMPHIDAE | |
| | *Gomphoides praevia* St. Quentin, 1967 | 1 |
| | LIBELLULIDAE | |
| | *Anatya guttata* Erichson in Schomburgk, 1848 | 6 |
| | *Anatya januaria* Ris, 1911 | 3 |
| | *Dasythemis essequiba* Ris, 1919 | 1 |
| | *Dasythemis venosa* Burmeister, 1839 | 1 |
| | *Diastatops obscura* Fabricius, 1775 | 2 |
| | *Diastatops nigra* Montgomery, 1940 | 9 |
| | *Elasmothemis alcebiadesi* Santos, 1945 | 6 |
| | *Elga leptostyla* Ris, 1909 | 1 |
| | *Erythemis carmelita* Williamson, 1923 | 1 |
| | *Erythrodiplax* sp2 Brauer, 1868 | 1 |
| | *Erythrodiplax* sp3 Brauer, 1868 | 1 |
| | *Erythrodiplax avittata* Borror,1942 | 1 |
| | *Erythrodiplax castanea* Burmeister, 1839 | 14 |
| | *Erythrodiplax famula* Erichson in Schomburgk, 1848 | 1 |
| | *Erythrodiplax fusca* Rambur, 1842 | 58 |
| | *Erythrodiplax latimaculata* Ris, 1911 | 1 |
| | *Erythrodiplax maculosa* Hagen, 1861 | 1 |
| | *Erythrodiplax media* Borror, 1942 | 3 |
| | *Erythrodiplax umbrata* Linnaeus, 1758 | 4 |
| | *Macrothemis tenuis* Hagen, 1868 | 4 |
| | *Micrathyria artemis* Ris, 1911 | 8 |
| | *Micrathyria catenata* Calvert, 1909 | 1 |
| | *Micrathyria ungulata* Förster, 1907 | 12 |
| | *Nephepeltia phryne* Perty, 1833 | 1 |
| | *Oligoclada umbricola* Borror, 1931 | 1 |
| | *Orthemis attenuata* Erichson in Schomburgk, 1848 | 3 |
| | *Orthemis discolor* Burmeister, 1839 | 4 |
| | *Perithemis thais* Kirby, 1889 | 17 |
| | *Uracis infumata* Rambur, 1842 | 2 |
| | Total Abundance | 689 |
| | Zygoptera Abundance | 520 |
| | Anisoptera Abundance | 169 |

**Table A4.** Genera recorded in the cacao-cabruca areas, in the streams sampled in the southern region of Bahia.

| Subordem | Family/Genera | Abundance Larvae |
|---|---|---|
| **Zygoptera** | | |
| | CALOPTERYGIDAE | |
| | *Hetaerina* | 26 |
| | COENAGRIONIDAE | |
| | *Argia* | 33 |
| | *Enallagma* | 2 |
| | *Homeoura* | 43 |
| | *Heliocharis* | 1 |
| | LESTIDAE | |
| | *Lestes* | 5 |
| | HETERAGRIONIDAE | |
| | *Heteragrion* | 76 |
| | *Dimeragrion* | 1 |
| | *Oxystigma* | 6 |
| | PERILESTIDAE | |
| | *Perilestes* | 30 |
| **Anisoptera** | | |
| | AESHNIDAE | |
| | *Roppaneura* | 4 |
| | *Castoraeschna* | 1 |
| | *Coryphaeschna* | 11 |
| | GOMPHIDAE | |
| | *Agriogomphus* | 36 |
| | *Gomphoides* | 2 |
| | *Epigomphus* | 72 |
| | *Peruviogomphus* | 1 |
| | *Phyllocycla* | 5 |
| | *Progomphus* | 4 |
| | LIBELLULIDAE | |
| | *Brechmorhoga* | 19 |
| | *Cannaphila* | 20 |
| | *Dasythemis* | 13 |
| | *Dythemis* | 24 |
| | *Elga* | 26 |
| | Erythemis | 19 |
| | *Erythrodiplax* | 6 |
| | *Gynothemis* | 11 |
| | *Macrothemis* | 22 |
| | *Micrathyria* | 19 |
| | *Orthemis* | 14 |
| | *Pachydiplax* | 16 |
| | *Perithemis* | 6 |
| | *Rhodopygia* | 16 |
| | *Tramea* | 7 |
| | Total Abundance | 647 |
| | Zygoptera Abundance | 229 |
| | Anisoptera Abundance | 418 |

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
