# Peer review of "Dragonflies (Odonata) in Cocoa Growing Areas in the Atlantic Forest: Taxonomic Diversity and Relationships with Environmental and Spatial Variables"

_diversity, doi:10.3390/d14110919_

Round 1

Reviewer 1 Report (Previous Reviewer 1)

In general I like to topic of the manuscripts

I would like to suggest to add (based on larvae and adult stage or two life stage...etc.) in the title but it is ok in its current state if you could not.

Author Response

Dear reviewer

We thank you for your time and dedication in reading and making suggestions for our manuscript.

Regarding the inclusion of larvae and adults in the title, in order not to make it even longer, we decided to keep it as it is.
We have added this information (adults and larvae) at the beginning of the abstract to make it clear to future readers of the our manuscript.

Best regards

Marciel e Láis

Reviewer 2 Report (New Reviewer)

I find you have done a great job and I hope these comments will help you improve it to make a great publication. I have two main concerns and some other minor comments:

1. In general the idea of forest-covered crops is in the paper but it is hard to get the effect of the crops compared to a natural forest. Why didn't you sample any conserved forest areas? that would act as a control and will allow you to assess the effect of cocoa crops. After the great sampling effort you have made I consider adding at least two more locations of conserved forest, without any crop around, to make a good contrast and clearly assess the effect of the crops.

2. Looking at the species you are recording is clear that lentic and lotic habitats are mixed in some of the sampled locations but not in all of them. It is known that lotic and lentic habitats harbor totally different odonate communities even if they are in the same area, so having both ecosystems in some areas will increase the richness independent of the forest coverage. This is a confusion factor you should untangle in the analysis, separating the areas with both aquatic ecosystems or adding the aquatic habitats as a factor in the analysis.

Also, it is not clear how are the water bodies embedded in the crops, are they in the borders, in the middle, are there ponds, lakes... what is the position of the water body in the crop is also important, maybe another confusion factor. Please describe this in the methods.

3. You are not giving the elevation of the areas sampled, are they at the same elevation? recent studies have shown that slight changes in elevation in tropical mountains can lead to a complete change in odonate communities.

Minor comments:

Please provide a better description of the sampling effort and collecting methods used.

Please provide a list of species per locality in a supplementary table, so the analysis can be redone.

Author Response

I find you have done a great job and I hope these comments will help you improve it to make a great publication. I have two main concerns and some other minor comments:

R: We thank you for your time and dedication in reading and making suggestions for our manuscript.

1. In general the idea of forest-covered crops is in the paper but it is hard to get the effect of the crops compared to a natural forest. Why didn't you sample any conserved forest areas? that would act as a control and will allow you to assess the effect of cocoa crops. After the great sampling effort you have made I consider adding at least two more locations of conserved forest, without any crop around, to make a good contrast and clearly assess the effect of the crops.

R: We appreciate your comment. However, we did not include data from conserved forest areas in the study because we already have work that compares different land uses with Odonata diversity, including pasture and cocoa plantation areas with preserved areas (reference 14). The data from this study are already used mainly in some parts of the introduction and mainly in the discussion of this article. It is he who provides the information that supports the discussions that cocoa-growing areas maintain a high diversity of species. Therefore, we think it would be too repetitive to include this comparison again and that it would be out of the scope of this study, which is to describe the diversity of Odonata in cocoa growing areas and the relationships of these assemblages with local and spatial variables.

The issues of comparisons raised can be seen throughout the text, for example, in the first paragraph of the discussion of our manuscript.

"Discussion
The results indicated that the cabruca areas maintain a vast diversity and richness of Odonata species (adults and larvae). Studies with the group in the Atlantic Forest that compare different land uses have highlighted the importance of cocoa cultivation areas for the diversity of species in the region [14]."

Reference
14.    Santos, LR; Rodrigues, ME. Land Uses for Pasture and Cacao Cultivation Modify the Odonata Assemblages in Atlantic Forest Areas. Diversity 2022, 14, 672 

2. Looking at the species you are recording is clear that lentic and lotic habitats are mixed in some of the sampled locations but not in all of them. It is known that lotic and lentic habitats harbor totally different odonate communities even if they are in the same area, so having both ecosystems in some areas will increase the richness independent of the forest coverage. This is a confusion factor you should untangle in the analysis, separating the areas with both aquatic ecosystems or adding the aquatic habitats as a factor in the analysis.

R:  Yes, we agree with you that the list features species recorded in lentic environments and we also agree that lentic and lotic environments harbor different dragonfly communities. However, in our study we did not sample any lentic environments, all lotic.
What studies have shown and what we have also discussed is that land use changes can bring some changes in lotic environments such as the decrease in flow caused mainly by soil erosion, which can lead to the formation of large backwater areas. And this factor, together with the increase in vegetation gaps, favors the establishment of species that are associated with lentic environments. Therefore, even sampling lotic environments we have records of these species.

Discussion (line 375)
In cabruca areas, changes in the amount forest cover, riparian vegetation, canopy openings, erosion of stream banks, increase the number of vascular plants in the channels and sediments in the stream, as well as the air and water temperature. These modifica-tions may favor the abundance and permanence of species considered open-area specialists and habitat generalists [33], as in the cases of Perithemis thais Kirby, 1889, Argia chapadae Calvert, 1909, Telebasis willinki Fraser, 1948 and Erythrodiplax latimaculata Ris, 1911, which have also been recorded in other studies of altered areas [25,26,32,55,56]. 

2.1 Also, it is not clear how are the water bodies embedded in the crops, are they in the borders, in the middle, are there ponds, lakes... what is the position of the water body in the crop is also important, maybe another confusion factor. Please describe this in the methods.

R: As mentioned, all sampled locations are lotic. These environments are mostly inserted in the middle of cocoa growing areas. And within the areas of cocoa growing areas there were no dams or lakes. We emphasize this in the manuscript methodology.

(Line 112) This study was conducted in six municipalities, namely Una, Buerarema, São José da Vitória, Ilhéus, Uruçuca, and Itacaré, which make up the southern region of the state of Bahia. The adult specimens were collected in 22 streams, 16 of which were also used to collect the larval specimens (Figure 1) (supplementary material, Appendix 01). The samples were collected in first to third-order streams from September to November 2019 and in July and August 2020. The selected properties for sampling belonged to organic cocoa producers of the Cooperativa Cabruca (an agricultural cooperative). In these areas, the cocoa plants are grown under the shade of native trees and in consortium with others crops, such as banana, açai berry, cupuaçu, vanilla, and palm oil. The properties are considered small with a cultivation system organic and without the use of pesticides. The channels of the lotic environments sampled were mostly located in the middle of the cocoa growing areas.

3. You are not giving the elevation of the areas sampled, are they at the same elevation? recent studies have shown that slight changes in elevation in tropical mountains can lead to a complete change in odonate communities.
R: All areas sampled are from municipalities that are located on the coast of the state of Bahia. None of the areas have a big difference in altitude. Therefore, we did not enter altitude as an analysis variable, the variation found is very small.

Minor comments:

Please provide a better description of the sampling effort and collecting methods used.

R: We have improved the description.

(Line 129) The adult specimens were collected with an entomological net, in segments of 100 meters on both banks of the streams using an entomological net and a sampling effort of 01:30 hour for each sampled stream between 9:00 am and 4:00 pm. Two sampling campaigns were carried out at each stream to increase the representativeness of the assemblages at each sampling site. 

The larvae were collected in areas of rapids and backwater in different habitats using a D-frame net and sieve in all compartments of the environment. A 30-meter segment on the shore was selected along each stream and 20 stretches of 1 meter were sampled in proportion to the number of registered habitats in each stream (sand, leaves, particulate organic matter, gravel, consolidated bedrock), according to the method of Barbour [37]. Active searches were performed using the sieve, for 15 minutes, in the different habitats within the 100-meter stretch in areas that had not been sampled with the D-frame net. 

Please provide a list of species per locality in a supplementary table, so the analysis can be redone.
R: We have added the table in the supplementary material.

Round 2

Reviewer 2 Report (New Reviewer)

I understand you have already made the comparison between forest and cabruca ecosystems, non the less it would be good to be more detailed in the discussion and show which forest species are not found in the cabruca, no matter how much vegetation cover it has.

Please provide axes with the values of the percentage of vegetation coverage for the graphs.

yours results are contrasting with recent studies in other tropical areas showing forest supports a large number of unique species (Novelo-Gutiérrez, R., Londoño, G. A., & Cordero-Rivera, A. (2021). The importance of tropical mountain forests for the conservation of Dragonfly biodiversity: a case from the Colombian Western Andes. International Journal of Odonatology24, 233-247.). It would be interesting to discuss this contrasting communities, why do you think the species you are finding in Bahia are not as specialized as the ones reported in this paper? maybe the rapid sampling is not allowing to find the more rare species?

You mentioned your analysis included a distance matrix to account for correlation between sites and possible pseudoreplication. You are not providing the coordinates of the sampled sites, so this analysis can not be performed again with the data you are sharing, which obscures your analysis, please provide the complete data.

Author Response

I understand you have already made the comparison between forest and cabruca ecosystems, non the less it would be good to be more detailed in the discussion and show which forest species are not found in the cabruca, no matter how much vegetation cover it has.

R: We added this information to the discussion.

(LINE 368) "However, it is worth mentioning that some species considered specialists in forests that have already been recorded in the region in areas of native forest in the Atlantic Forest [14], were not recorded in the cacao cultivation areas sampled. For example, Aceratobasis cornicauda (Calvert, 1909), Heliocharis amazona Selys, 1853, Kiautagrion acutum Santos, 1961 and Leptagrion macrurum (Burmeister, 1839), the last two species being phytotelmata [14]. Emphasizing that even crops that maintain a part of the native vegetation such as cabruca areas may not maintain some groups of species considered more sensitive or with more specialized habitats."

Please provide axes with the values of the percentage of vegetation coverage for the graphs.

R: We enter the percentage of forest cover on the axes of the graphs. The values for each sampling site can be consulted in the Supplementary material - Appendix 1.

yours results are contrasting with recent studies in other tropical areas showing forest supports a large number of unique species (Novelo-Gutiérrez, R., Londoño, G. A., & Cordero-Rivera, A. (2021). The importance of tropical mountain forests for the conservation of Dragonfly biodiversity: a case from the Colombian Western Andes. International Journal of Odonatology, 24, 233-247.). It would be interesting to discuss this contrasting communities, why do you think the species you are finding in Bahia are not as specialized as the ones reported in this paper? maybe the rapid sampling is not allowing to find the more rare species?

R: The work mentioned is very interesting. However, our data is from areas that are no longer native forests. These are areas that have already suffered some degree of impact for cocoa cultivation. For this reason, we don't have as many species specialized in native forest environments in these areas.

Different from the results of Novelo-Gutiérrez et al 2021 that evaluated an area with very specific characteristics (mountains) in an altitude gradient. Natural environmental factors such as altitude for example select groups of more specialized species. Therefore, the authors find a greater number of endemic and/or specialist species.

We carried out two sampling campaigns at each site at different times of the year. The lack of more specialist species from forested areas may simply be associated with the modification of areas for cultivation. As much as this type of cultivation and modification of land use is not very aggressive, a part of the most sensitive species can disappear from these areas. Therefore, they are not being found. We have added this information to the discussion (line 368).

You mentioned your analysis included a distance matrix to account for correlation between sites and possible pseudoreplication. You are not providing the coordinates of the sampled sites, so this analysis can not be performed again with the data you are sharing, which obscures your analysis, please provide the complete data.

R:  The coordinates of the sampled sites are in the supplementary material - Appendix 1.

This manuscript is a resubmission of an earlier submission. The following is a list of the peer review reports and author responses from that submission.

Round 1

Reviewer 1 Report

This is an interesting and well-designed study with a good detailed output.

The results will help in evaluating and to understand the effects of agroforestry systems on Odonata biodiversity, and maybe as baseline for other groups of aquatic insects.

I have minor comments that, if considered, will improve the text.

Methods:

I am very interested to see in the Methods section detailed description of HII of your sites if you could, even in small table or you could add it in supplementary material.

Results

- I would like to see and recommend to add one more result about the difference in the response between Zygoptera and Anisoptera (as a different groups) in such work because I believe that they have different sensitivity.

- The order of Figures? There are only five figures not six

- Add the (t) to adult in Y axis legend in Figure 3

The discussion is well written

Author Response

Comments and Suggestions for Authors

This is an interesting and well-designed study with a good detailed output.

The results will help in evaluating and to understand the effects of agroforestry systems on Odonata biodiversity, and maybe as baseline for other groups of aquatic insects.

I have minor comments that, if considered, will improve the text.

Methods:

I am very interested to see in the Methods section detailed description of HII of your sites if you could, even in small table or you could add it in supplementary material.

R: We made a table with the HII values of each sample site and placed it as supplementary material for the study. (See supplementary material Appendix 01)

Results

- I would like to see and recommend to add one more result about the difference in the response between Zygoptera and Anisoptera (as a different groups) in such work because I believe that they have different sensitivity.

R: We appreciate the suggestion. However, the responses between the suborders can be evaluated in the analyzes through the species of each group. The analysis of the results brings the relationship of the species with the HII and with the environmental variables. Therefore, we find it redundant and uninformative for this comparison to be made at a taxonomic level as high as a suborder.

- The order of Figures? There are only five figures not six

R: I think there was a mistake. I already confirmed in ms and there are 6 figures.

- Add the (t) to adult in Y axis legend in Figure 3

R: Ok, done.

The discussion is well written

R: Thanks for the comments and compliments.

Reviewer 2 Report

General comment.

Please note that I’m not a native English speaker so I can hardly make comments about the appropriate use of the English language in the paper.

Regarding the scientific interest of the paper, the authors statement that the “cabruca” agrosystem “maintain a great diversity of dragonflies” or “cabruca areas maintain a vast diversity and richness of Odonata species” is not clearly established and difficult to appreciate for the reader in the actual version of the paper. As the authors provide no direct comparison with either habitats of pristine conditions and/or more intensive/degraded agro-systems that we may expect to be more deleterious (such as pasture, soja fields, orange orchards...), the reader can hardly understand how “great” or “vast” is the diversity of Odonata hosted in the cabruca.

There are numerous table and figures showing the results of quite complicated statistical analysis regarding the habitat characteristics but a basic comparison of species richness, % of specialist species, % of endemic species between cabruca and other habitats is lacking. I’am afraid this cannot be solved a posteriori.

Also, I’m not completely at ease with such statistical analysis but I can hardly agree with the authors when they wrote “According to these results, the different life stages depend on the same environmental characteristics”. Presence of adults and nymphs are not independant but it does not mean that each life stage depend on the same environmental characteristics. 

Detailed comment.

Line 16 : Please specify « nymphs » rather than “immature individuals” as “immature individuals” may be confusing (because of the Odonata life-cycle with adults being “immature” right after the emergence and then requiring a “maturation” time to become “sexually mature”).

Line 30: I think something is missing in the sentence ending by “…along with the associated [1-3].”

Line 46: “Theobroma cacao » should be in italic police and the author of this species and date of first description should be mentioned: “Theobroma cacao L. 1753”.

Line 47: I feel uncomfortable with such sentence “In addition to the economic impact of this type of cultivation on the region, it has contributed to the conservation of part of the region’s flora and biodiversity”. Regarding to the initial state of biodiversity, we can assume that it has contributed to a degradation of biodiversity. Probably at a lesser extent than if those lands would have become soja fields or pastures, but I would not say that it has contributed to the conservation of part of autochtonous fauna and flora diversity.

Line 47: “of the region’s flora and biodiversity” is inappropriate as flora is not apart from biodiversity but included in. Please prefer “of the region flora and fauna diversity” or “of the region biodiversity”.

Line 49: The “cabruca” agro-system would deserve a bit more detailed description. It seems that the shading trees are not always natives as growers also use fruit trees as shading trees. Also, even if the tree species are natives, they are selected by growers for their economic value and thus their diversity is probably much lower than the diversity previously hosted by those lands.

Line 60: what is the “ciliary vegetation”?

Line 68, 69, 70: It is somehow contradictory to write that Odonata are great bioindicators and highly tolerant to changes in the natural environment.

Line 72: I would not use the term “immature individuals” but rather refer to “larvae” or “nymphs”

Line 81-82: I do not understand why the autors refer to “the species”

Line 88 and 94 to 98: The authors should announce some question or present hypothesis to be tested rather than yet evoking the results.

Line 101 to 103: Please refer to figure 1.

Line 104: please indicate if you are specifically talking about the odonatofauna or about biodiversity in general.

Line 111: Precise coordinates of each sampling sites should be given in a table as supplementary material.

Line 124: More details should be provided in the “sample collection” section, such as the precise sampling periods (dates of beginning and end of the first and second sampling periods), the precise conditions of adult and nymph preservation…

Figure 1: The map should contain more information. For example, I could not easily check if your sites were close or far from the sites studied by Brasil et al. 2016.

Please add a map of Brazil. Please add on the central map the names of the localities you cited in the text (Una, Buerarema, São José da Vitória, Ilhéus, Uruçuca, and Itacaré); please also increase the size of the front used for your legends (there is enough empty space below the left map to use a larger front and make sure it will be easily readable). In connection with line 111 and 112, please make sure the reader will be able to locate the sites were both larvae and adults were sampled versus the site were only adults were sampled (use different colors for your circles or circles versus squares). You should also associate each sampling point to a number for identification.

Line 116: please be more precise than “the use of pesticide is avoided”. What do you mean?

Line 136: I think “…identification based on some keys according to [23,38-42], among others” should be rephrased. My suggestion would be: “…identification based on available keys [23,38-42]”,

Line145: The methods to obtain HII is not very clear. Please specify “as proposed by Nessimian [42] for Central Amazonian streams”. Please replace “12 questions” by “12 habitat characteristics”. Please make clear what is the “buffer zone” you refer to. It should be clear that the Nessimian HII is a qualitative (not quantitative) evaluation with each habitat characteristic being scored. I would not use the term “metrics” for the qualitative habitat characteristics.

Line 153: if the HII have been widely used in ecological studies, please provide references of those studies, not only the reference of the seminal paper.

Line 174: Please list the physicochemical parameters of the water that were measured together with temperature, pH, conductivity, dissolved oxygen, and salinity.

Line 177: Table 1 is missing when searching with pdf search tool (so please avoid to name “Table 01” except if required by the editor).

Line 217: Reference to table 1 is an error (rather refer to “Appendix 1”).

Line 218: Scientific names should be given with their authors and year of first description.

Line 221: Please replace “immature individuals » by nymphs or larvae.

Figure 3: readers should be able to link each column to a sampling point in figure 1 so please identify each column. There are 22 columns while you previously mentioned that you sampled odonata on 21 sites. It should be clearly stated that the upper line stands for the integrity index.

Also,, you could add for each sampling point the total number of individual. Please replace IIH by HII.

Line 237: Please replace “immature individuals” by “larvae” or “nymphs”

Line 250: Please replace “immature individuals” by “larvae” or “nymphs”.

Figure 4. I cannot understand how you can have in this figure nine sampling point with high HII (value seems to be “1”) while in figure 3 only 3 sites reached an HII value of around 0,8. As the sampling points for larvae are a subset of the sampling points for adults it would not be possible. Please also replace IIH by HII.

Line 262: Please replace “immature individuals” by “larvae” or “nymphs”

Table 2: Please replace the title of the 3rd column “Axles” by “Axes”

Line 305: Please replace “immature individuals” by “larvae” or “nymphs”

Line 325: Please provide the references showing that Perilestes fragilis, Aceratobasis mac, ilenta, Forcepsioneura serrabonita and Epigomphus paludosus are specialists of forests. It also would be valuable to compare the species richness observed in cabruca with the regional species pool and clearly state what % of species could be shown to be hosted in the cabruca agro-system?

Line 331: Please replace IIH by HII

Line 339: This sentence should be rephrased “species had relatively abundant distributions along the entire gradient of the HII” because “relatively abundant distributions” has no sense.

Line 353-354: “According to these results, the different life stages depend on the same environmental characteristics, and the changes in these variables can modify the structuring of local assemblages in cabruca areas”. These This should be rephrased. There is no demonstration that larvae and adults depend on the same environmental variables: it is quite normal to find larvae where adults are present (to mate, to lay eggs…), thus under the same environmental conditions.

Line 360: What do you mean by “destabilization of the margins”

Appendix 1. Please replace “cacau” by “cacao”. Also “Calopterygidae” should not be in the same case as “Family/species. Please delete “Cabruca” from the top of the last column which should be “abundance” only. I would suggest to clearly identify which species are considered as generalist versus forest specialist and open-area specialists in this table. Replace “Telebasis corollina » by “Telebasis corallina

Appendix 2. Please sort the genera by families as you did for the species in the appendix 1.

Table 1. Number of replicates should be mentioned.

Line 608: I did not check the reference list in details however please check the Diversity guidelines for the reference list. I think you should not use capital letters at the beginning of each word of the reference title.

Line 612: I did not check the reference list in details however please check the Diversity guidelines for the reference list. It seems that sometimes you cite the full name of the journals and sometimes its short version (Agroforest Syst is the short name of Agroforestry Systems).

Author Response

------------------------------------------------------------

Comments and Suggestions for Authors

General comment.

Please note that I’m not a native English speaker so I can hardly make comments about the appropriate use of the English language in the paper.

Regarding the scientific interest of the paper, the authors statement that the “cabruca” agrosystem “maintain a great diversity of dragonflies” or “cabruca areas maintain a vast diversity and richness of Odonata species” is not clearly established and difficult to appreciate for the reader in the actual version of the paper. As the authors provide no direct comparison with either habitats of pristine conditions and/or more intensive/degraded agro-systems that we may expect to be more deleterious (such as pasture, soja fields, orange orchards...), the reader can hardly understand how “great” or “vast” is the diversity of Odonata hosted in the cabruca.

There are numerous table and figures showing the results of quite complicated statistical analysis regarding the habitat characteristics but a basic comparison of species richness, % of specialist species, % of endemic species between cabruca and other habitats is lacking. I’am afraid this cannot be solved a posteriori.

Also, I’m not completely at ease with such statistical analysis but I can hardly agree with the authors when they wrote “According to these results, the different life stages depend on the same environmental characteristics”. Presence of adults and nymphs are not independant but it does not mean that each life stage depend on the same environmental characteristics.

R: We are grateful for the reviewer's comments and review. We respond to all comments peer-to-peer.

However, regarding general comments, we would like to inform you that we have added in the references and in the text discussion the work of Santos and Rodrigues (2022) (See reference below), which discusses the comparison of the richness of the cocoa growing areas with others. uses such as pasture and with native areas.

Santos, L.R.; Rodrigues, M.E. Land Uses for Pasture and Cacao Cultivation Modify the Odonata Assemblages in Atlantic Forest Areas. Diversity 2022, 14, 672 https://doi.org/10.3390/d14080672 

We do not agree with the reviewer on the last general comment. For the vast majority of Odonata species, adults choose environmental characteristics for egg laying and larval development (Corbet 1999, Rodrigues et al., 2018). And therefore, as our data have shown, some environmental variables can indeed be shared between adults and larvae.

Corbet, P. S. 1999. Dragonflies: Behavior and Ecology of Odonata. Comstock Publ. Assoc., Ithaca, NY. 829 p.

Rodrigues ME, Roque FO, Ferreira RGN, Saito VS, Samways MJ (2018) Egg-laying traits reflect shifts in dragonfly assemblages in response to different amount of tropical forest cover. Insect Conserv  Diver, 11: 01-10 https://doi.org 10.1111/icad.12319

Detailed comment.

Line 16 : Please specify « nymphs » rather than “immature individuals” as “immature individuals” may be confusing (because of the Odonata life-cycle with adults being “immature” right after the emergence and then requiring a “maturation” time to become “sexually mature”).

R: We have modified the term immature by larvae throughout the text.

Line 30: I think something is missing in the sentence ending by “…along with the associated [1-3].”

 R: We corrected the sentence.

“…along with the biodiversity associated [1-3].”

Line 46: “Theobroma cacao » should be in italic police and the author of this species and date of first description should be mentioned: “Theobroma cacao L. 1753”.

R: Ok, add.

Line 47: I feel uncomfortable with such sentence “In addition to the economic impact of this type of cultivation on the region, it has contributed to the conservation of part of the region’s flora and biodiversity”. Regarding to the initial state of biodiversity, we can assume that it has contributed to a degradation of biodiversity. Probably at a lesser extent than if those lands would have become soja fields or pastures, but I would not say that it has contributed to the conservation of part of autochtonous fauna and flora diversity.

 A: We agreed with the reviewer and modified the way of writing the sentence.

This agroforestry system is considered favorable for the conservation of natural resources and local biodiversity, since it maintains part of the native forest structure and, consequently, protects terrestrial ecosystems, sustains part of ecological services, and maintains the species diversity of fauna and flora [12-15].

Line 47: “of the region’s flora and biodiversity” is inappropriate as flora is not apart from biodiversity but included in. Please prefer “of the region flora and fauna diversity” or “of the region biodiversity”.

R: We corrected the sentence.

Line 49: The “cabruca” agro-system would deserve a bit more detailed description. It seems that the shading trees are not always natives as growers also use fruit trees as shading trees. Also, even if the tree species are natives, they are selected by growers for their economic value and thus their diversity is probably much lower than the diversity previously hosted by those lands.

 R: We add this information in the topic (Study area) line 126

In these areas, the cocoa plants are grown under the shade of native trees and in consortium with others crops, such as banana, açai berry, cupuaçu, vanilla, and palm oil. The properties are considered small with a cultivation system organic and without the use of pesticides.

Line 60: what is the “ciliary vegetation”?

 R: We replaced the term with "riparian vegetation"

Line 68, 69, 70: It is somehow contradictory to write that Odonata are great bioindicators and highly tolerant to changes in the natural environment

R: We corrected the sentence.

……This is because there are species are more or less tolerant to changes in the natural environment.

Line 72: I would not use the term “immature individuals” but rather refer to “larvae” or “nymphs”

 R: We have modified the term immature by larvae throughout the text.

Line 81-82: I do not understand why the autors refer to “the species”

 R: We replaced the term with the word group.

Line 88 and 94 to 98: The authors should announce some question or present hypothesis to be tested rather than yet evoking the results.

R: We have modified the sentence

“Our prediction is that.........

Line 101 to 103: Please refer to figure 1.

 R: Ok, done.

Line 104: please indicate if you are specifically talking about the odonatofauna or about biodiversity in general.

  R: Ok, done.

Line 111: Precise coordinates of each sampling sites should be given in a table as supplementary material.

R: We add a table with coordinates as supplementary material, see Appendix 01. 

Line 124: More details should be provided in the “sample collection” section, such as the precise sampling periods (dates of beginning and end of the first and second sampling periods), the precise conditions of adult and nymph preservation…

R: Information on collection dates is already described in the study area section. And the information on specimen preservation conditions is unnecessary.

Figure 1: The map should contain more information. For example, I could not easily check if your sites were close or far from the sites studied by Brasil et al. 2016.

R: The aforementioned researcher (Brasil) works in areas of the Amazon and therefore the data collected by him no relationship with our study.

Please add Please add on the central map the names of the localities you cited in the text (Una, Buerarema, São José da Vitória, Ilhéus, Uruçuca, and Itacaré); please also increase the size of the front used for your legends (there is enough empty space below the left map to use a larger front and make sure it will be easily readable). In connection with line 111 and 112, please make sure the reader will be able to locate the sites were both larvae and adults were sampled versus the site were only adults were sampled (use different colors for your circles or circles versus squares). You should also associate each sampling point to a number for identification.

 R: We redid the map by adding a map of Brazil and the names of the localities you cited in the text (Una, Buerarema, São José da Vitória, Ilhéus, Uruçuca, and Itacaré).

About adult and larval sampling sites we put this information in a table. See supplementary material, appendix 01.

Line 116: please be more precise than “the use of pesticide is avoided”. What do you mean?

 R: Cultivation system are organics and without the use of pesticides. We rewrite the sentence.

Line 136: I think “…identification based on some keys according to [23,38-42], among others” should be rephrased. My suggestion would be: “…identification based on available keys [23,38-42]”,

 R: Ok, done.

Line145: The methods to obtain HII is not very clear. Please specify “as proposed by Nessimian [42] for Central Amazonian streams”. Please replace “12 questions” by “12 habitat characteristics”. Please make clear what is the “buffer zone” you refer to. It should be clear that the Nessimian HII is a qualitative (not quantitative) evaluation with each habitat characteristic being scored. I would not use the term “metrics” for the qualitative habitat characteristics.

 R: We made the suggested changes

“12 questions” by “12 habitat characteristics”.

and

“metrics” for “the qualitative habitat characteristics”

And we deleted the term “buffer zone” from the sentence.

Line 153: if the HII have been widely used in ecological studies, please provide references of those studies, not only the reference of the seminal paper.

R: The reference cited at the end of the paragraph is from a meta-analysis study that evaluated and cites many articles using the HII. Therefore we find it unnecessary to add more references in the text.

  • Brasil, L.S., Lima, E.L., Spigoloni, Z.A., Ribeiro-Brasil, D.R.G., Juen, L. The habitat integrity index and aquatic insect communities in tropical streams: a meta-analysis. Ecological Indicators 2020 116:0–2. https://doi.org/10.1016/j.ecolind.2020.106495, 106495

Line 174: Please list the physicochemical parameters of the water that were measured together with temperature, pH, conductivity, dissolved oxygen, and salinity.

 R: Ok, done.

Line 177: Table 1 is missing when searching with pdf search tool (so please avoid to name “Table 01” except if required by the editor).

 R: Ok, done.

Line 217: Reference to table 1 is an error (rather refer to “Appendix 1”).

 R: Adjusted.

Line 218: Scientific names should be given with their authors and year of first description.

  R: Ok, done.

Line 221: Please replace “immature individuals » by nymphs or larvae.

 R: Ok, done.

Figure 3: readers should be able to link each column to a sampling point in figure 1 so please identify each column. There are 22 columns while you previously mentioned that you sampled odonata on 21 sites. It should be clearly stated that the upper line stands for the integrity index.

Also,, you could add for each sampling point the total number of individual. Please replace IIH by HII.

 R: We redid figure 3 and corrected all the points raised. There are 22 bars because there are 22 sampling sites and not 21. Now we show that the upper line represents the HII. And finally we replace IIH with HII.

Line 237: Please replace “immature individuals” by “larvae” or “nymphs”

 R: Ok, done.

Line 250: Please replace “immature individuals” by “larvae” or “nymphs”.

 R: Ok, done.

Figure 4. I cannot understand how you can have in this figure nine sampling point with high HII (value seems to be “1”) while in figure 3 only 3 sites reached an HII value of around 0,8. As the sampling points for larvae are a subset of the sampling points for adults it would not be possible. Please also replace IIH by HII.

 R: We redid figure 4 and corrected all the points raised. Now the bars representing the HII are in accordance with the bars in figure 3. We also show that the upper line represents the HII. And finally we replace IIH with HII.

Line 262: Please replace “immature individuals” by “larvae” or “nymphs”

 R: Ok, done.

Table 2: Please replace the title of the 3rd column “Axles” by “Axes”

 R: Ok, done.

Line 305: Please replace “immature individuals” by “larvae” or “nymphs”

R: Ok, done.

Line 325: Please provide the references showing that Perilestes fragilis, Aceratobasis mac,ilenta, Forcepsioneura serrabonita and Epigomphus paludosus are specialists of forests. It also would be valuable to compare the species richness observed in cabruca with the regional species pool and clearly state what % of species could be shown to be hosted in the cabruca agro-system?

 R: This information is already cited in the discussion of the article.

(Line 355) The relative abundance of some adult species and genera of larvae was associated with areas with higher HII values, as in the cases of the species Aceratobasis macilenta (Rambur, 1842)  and Hetaerina longipes Hagen in Selys, 1853, usually considered forest specialist species, and the genera Agriogomphus and Peruviogomphus and species Epigomphus paludosus Hagen in Selys, 1854, composed of more sensitive individuals that need greater environmental integrity and conditions more comparable to forest areas [14,53,54].

(Line 389) …which can help ensure the permanence of more sensitive species such as forest specialists. These forest specialists include Forcepsioneura serrabonita Pinto & Kompier, 2018 and Perilestes fragilis Hagen in Selys, 1862, as well as species of the genera Heteragrion, Peruviogomphus and Agriogomphus [14,26,56,60-63].

we would like to inform you that we have added in the references and in the text discussion the work of Santos and Rodrigues (2022) (See reference below), which discusses the comparison of the richness of the cocoa growing areas with others. uses such as pasture and with native areas.

Santos, L.R.; Rodrigues, M.E. Land Uses for Pasture and Cacao Cultivation Modify the Odonata Assemblages in Atlantic Forest Areas. Diversity 2022, 14, 672 https://doi.org/10.3390/d14080672 

Line 331: Please replace IIH by HII

 R: OK, Done.

Line 339: This sentence should be rephrased “species had relatively abundant distributions along the entire gradient of the HII” because “relatively abundant distributions” has no sense.

 R: we correct the sentence.

“A broad range of species had your distributions along the entire gradient of the HII…..”

Line 353-354: “According to these results, the different life stages depend on the same environmental characteristics, and the changes in these variables can modify the structuring of local assemblages in cabruca areas”. These This should be rephrased. There is no demonstration that larvae and adults depend on the same environmental variables: it is quite normal to find larvae where adults are present (to mate, to lay eggs…), thus under the same environmental conditions.

R: We do not agree with the reviewer on the last general comment. For the vast majority of Odonata species, adults choose environmental characteristics for egg laying and larval development (Corbet 1999, Rodrigues et al., 2018). And therefore, as our data have shown, some environmental variables can indeed be shared between adults and larvae.

Corbet, P. S. 1999. Dragonflies: Behavior and Ecology of Odonata. Comstock Publ. Assoc., Ithaca, NY. 829 p.

Rodrigues ME, Roque FO, Ferreira RGN, Saito VS, Samways MJ (2018) Egg-laying traits reflect shifts in dragonfly assemblages in response to different amount of tropical forest cover. Insect Conserv  Diver, 11: 01-10 https://doi.org 10.1111/icad.12319

Line 360: What do you mean by “destabilization of the margins”

 R: erosion of stream banks, we modify the sentence.

Appendix 1. Please replace “cacau” by “cacao”. Also “Calopterygidae” should not be in the same case as “Family/species. Please delete “Cabruca” from the top of the last column which should be “abundance” only. I would suggest to clearly identify which species are considered as generalist versus forest specialist and open-area specialists in this table. Replace “Telebasis corollina » by “Telebasis corallina

 R: Ok, done.

Appendix 2. Please sort the genera by families as you did for the species in the appendix 1.

 R: Ok, done.

Table 1. Number of replicates should be mentioned.

 R: We add this information to the table. See Appendix 02.

Line 608: I did not check the reference list in details however please check the Diversity guidelines for the reference list. I think you should not use capital letters at the beginning of each word of the reference title.

 R: Ok, done.

Line 612: I did not check the reference list in details however please check the Diversity guidelines for the reference list. It seems that sometimes you cite the full name of the journals and sometimes its short version (Agroforest Syst is the short name of Agroforestry Systems).

R: Ok, we check the reference list.
